# Pyramidal Solar Stills via Hollow Cylindrical Perforated Fins, Inclined Rectangular Perforated Fins, and Nanocomposites: An Experimental Investigation

Suha A. Mohammed [1], Ali Basem [2], Zakaria M. Omara [3], Wissam H. Alawee [4], Hayder A. Dhahad [1], Fadl A. Essa [3], Abdekader S. Abdullah [5,6], Hasan Sh. Majdi [7], Iqbal Alshalal [8], Wan Nor Roslam Wan Isahak [9] and Ahmed A. Al-Amiery [9,10,*]

1 Mechanical Engineering Department, University of Technology, Baghdad 10066, Iraq
2 Air Conditioning Engineering Department, Faculty of Engineering, Warith Al-Anbiyaa University, Karbala 56001, Iraq
3 Mechanical Engineering Department, Faculty of Engineering, Kafrelsheikh University, Kafrelsheikh 33511, Egypt
4 Control and Systems Engineering Department, University of Technology, Baghdad 10066, Iraq
5 Mechanical Engineering Department, College of Engineering, Prince Sattam Bin Abdulaziz University, Al-Kharj 16278, Saudi Arabia
6 Faculty of Engineering, Tanta University, Tanta 31527, Egypt
7 Department of Chemical Engineering and Petroleum Industries, Al-Mustaqbal University College, Babylon 51001, Iraq
8 Training and Workshops Center, University of Technology, Baghdad 10066, Iraq
9 Department of Chemical and Process Engineering, Faculty of Engineering and Built Environment, Universiti Kebangsaan Malaysia, Bangi 43600, Selangor, Malaysia
10 Energy and Renewable Energies Technology Center, University of Technology, Baghdad 10066, Iraq
* Correspondence: dr.ahmed1975@gmail.com or dr.ahmed1975@ukm.edu.my

**Abstract:** A practical study was conducted to improve the performance of conventional pyramidal solar stills (CPSS) using two types of fins with differing geometries, as well as nanocomposites of $TiO_2$ and graphene. The first fin was hollow, cylindrical, and perforated (HCPF), whereas the second fin was an inclined perforated rectangular fin (IPRF). The fins were integrated with the base of a solar still to evaluate their performance in comparison with a CPSS. The obtained experimental results demonstrated that the pyramidal solar still with hollow perforated cylindrical fins (PSS-HCPF) and the pyramidal solar still with inclined perforated rectangular fins (PSS-IPRF) produced more distillate than the PSS-HCPF and CPSS under all examined conditions. The daily productivities of the CPSS, PSS-HCPF, and PSS-IPRF were 3718, 4840, and 5750 mL/m², respectively, with the PSS-HCPF and PSS-IPRF improving the productivity by 31.3% and 55.9%, respectively, compared to that of the CPSS. In addition, using nanocomposites with PSS-IPRF improved the daily distillate production by 82.1%.

**Keywords:** perforated fins; pyramidal solar still; rectangular fins; nanocomposite; graphene

## 1. Introduction

Potable water is necessary for the survival of life on this planet; however, due to increasing global population, the demand for safe water for drinking and various applications is increasing [1]. In addition, the sustainability of energy and food sources is a major goal of all countries [2–4] associated with artificial intelligence applications [5]; therefore, there is a considerable demand for the use of renewable energy sources [6]. Many regions in the world, especially in remote areas and far from cities, suffer from energy shortages but have an abundance of solar energy throughout the year [7]. Solar distillation is one of the easiest and most cost-effective ways to obtain fresh water from salt water [8–10].

Many studies have assessed the performance of various solar still designs [11–13], such as single- and double-slope solar stills [14–17], conical solar stills [18,19], inclined

solar stills [20], tray solar stills [21], spherical and hemispherical solar stills [22,23], pyramidal solar stills [24,25], stepped solar stills [26,27], tubular solar stills [28–30], wick solar stills [31–34], solar stills with quantum dot nanofluids [35,36], dish solar stills [37], and multibasin solar stills [38–41], as well as factors affecting the performance of solar stills, including climatic or design factors [42]. Several techniques have been developed to increase the productivity of solar stills with various designs, including the use of primary solar heating in different ways to increase the evaporation rate [43,44]. The use of internal and external reflectors to reflect solar rays increases the heating of the absorption basin [45,46], and the use of fins in the absorption tank increases the heat exchange process between the absorption tank and water [47–49]. Different types of fillings can accelerate the evaporation process [50,51], and nanomaterials can be used to improve thermal performance [52–54]. The condensation process [52] can be improved using external and internal condensers [55–57], and phase-changing materials contribute to production in the absence of solar radiation [58].

The use of finned absorber plates can also improve the performance and increase the productivity of solar stills. Fins improve the thermal performance between the absorption plate and the water, which increases the temperature difference between the condensing surface and the water, thus increasing the productivity of the solar still. Several researchers have studied the use of fins in the distillation basin [59,60]. For instance, Alawee et al. [61] fixed a parallel plate with wick cords inside a pyramid solar still to increase productivity by 176%, whereas the redesigned solar system's thermal efficiency remained at 60.4%. Titanium oxide nanoparticles have also been used in several applications, such as photovoltaic panels [62] and antireflection applications [63,64]. Additionally, Velmurugan et al. [65] studied the effect of fins, bushings, and sponges on the performance of single-slope solar stills under varied operating conditions, showing that productivity increased by 45.5%, 29.6%, and 15.3% when using fins, wadding, and sponges, respectively. Ramadan et al. [66] studied the effect of fin geometry on the performance of single inclined solar stills, reporting increased productivity with increased fin height but decreased productivity with increased fin thickness. Moreover, increasing the number of fins reduced the daily productivity due to the increase in the shaded area and part of the solar radiation being blocked. Gnanaraj and Velmurugan [67] improved the productivity of double inclined solar stills by 58.4%, 69.8%, and 42.3% when using fins, black granite, and wicks, respectively, compared with a conventional solar still. Additionally, Panchal and colleagues [68] used a manganese oxide nanoparticle-coated absorber to enhance solar still output productivity by 19.5%. Rabhi et al. [69] also improved the performance of solar stills by using a finned and condenser absorption basin. A 15% improvement in performance was obtained using a finned absorber plate, and a 32% improvement was achieved using a condenser compared to a conventional solar still. Jani and Modi [70] increased the performance of a double inclined solar still by 54.2% using circular fins and by 26.8% using square sectional fins. Furthermore, Alawee et al. [71] increased the productivity of twin inclined solar stills combined with inclined rectangular perforated fins by 16–54% using a finned plate compared to a conventional flat absorbent plate.

However, the effects of using hollow cylindrical perforated fins and inclined rectangular perforated fins on pyramid solar still performance have not been investigated. Therefore, in this study, we applied hollow cylindrical perforated fins and inclined rectangular perforated fins to increase the surface area of the absorber (the still's base) and the rate of heat transmission between the saline water and the absorber to increase the productivity of solar basin stills. The novelty of this study is as follows:

1. We investigated the effect of adding hollow cylindrical perforated fins and inclined rectangular perforated fins on pyramid solar still performance;
2. We assessed of the effect of the number of fins on pyramid solar still performance;
3. We evaluated the influence of using graphene and titanium oxide ($TiO_2$) composite nanoparticles with saline water on pyramid solar still performance.

## 2. Materials and Methods

### 2.1. Fabrication of Solar Stills

The water desalination system shown in Figure 1 consisted of two solar stills of the same design and dimensions, a small tank for water supply, a network of pipes, a measuring jar, thermocouples, etc. One solar still was used as a reference distillate to study the effect of the proposed modifications to the second still, including the use of two types of geometrically shaped fins with nanomaterials. Both stills were made of galvanised iron, with a thickness of 1.5 mm and outside dimensions of 70 × 70 × 15 cm.

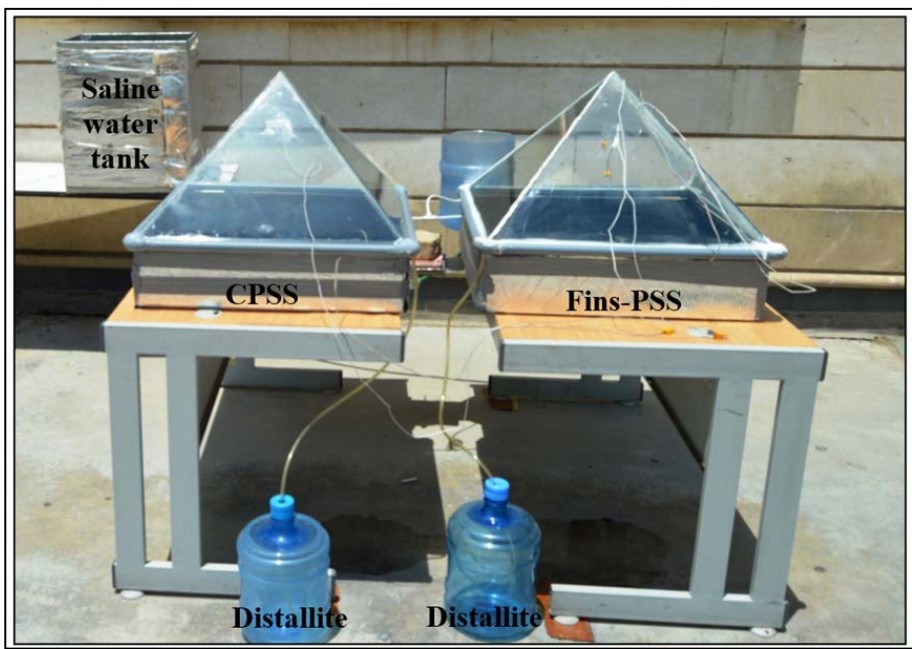

**Figure 1.** Experimental setup.

Four triangular sheets of 3 mm thick glass were used to cover the solar still and were tilted 30° from the horizon. Four channels were installed at the edge of the still basin to collect the condensed water from the inner surface of the glass toward the measuring jar. The glass panels were fixed tightly from the top and bottom using silicon material to avoid breakage and prevent steam from leaking. The basin still, inner walls, and fins were painted matte black to increase absorbency. The base was insulated using 5 cm thick fibreglass to reduce heat loss from the distillate to the outside environment. The capacity of the main tank supplying water to the solar still was 50 L, and the condensed water was collected through a measuring jar at the bottom of each solar still. The basin water depth was kept constant during all experiments at 3 cm.

Using finned plates in the base of the solar still increases the solar still basin surface area, thereby increasing the heat transfer between the base of the distiller and the water and increasing the water temperature. This leads to an increase in the difference between the water temperature and the glass cover, thereby increasing freshwater production.

Two types of finned plates with different geometric shapes were used with the modified pyramid solar stills. The first type was a hollow cylindrical fin with a diameter of 2 cm and a length of 3 cm, as shown in Figure 2. Thirty-two 2 mm diameter holes were positioned on the circumference and along the length of the cylinder. Figure 2 shows the 3D view of the hollow cylindrical fin of inline holes of θ = 90. The other fin was a rectangular, perforated fin inclined to the horizon at an angle of 45° (Figure 3). The width, thickness, and length of each fin are 3 cm, 0.15 cm, and 4.2 cm, respectively. The fins were designed to be tilted at an angle of 45° from the horizon line, which is the optimum angle of inclination for Baghdad city, Iraq, in winter.

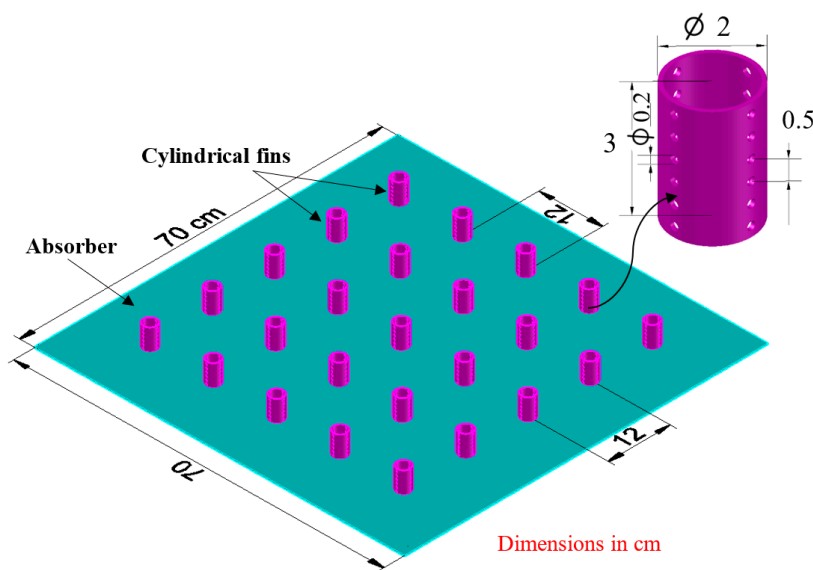

**Figure 2.** Pyramid solar still absorber with hollow perforated cylindrical fins.

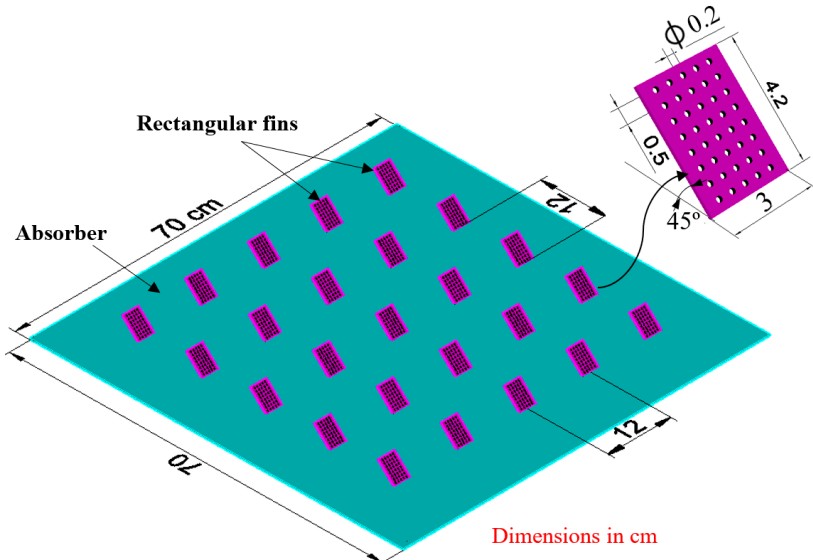

**Figure 3.** Pyramid solar still absorber with inclined perforated rectangular fins.

The number of fins was varied to study the effect on productivity. Three absorbent plates were manufactured for each modified solar still, PSS-IPRF and PSS-HCPF. Each absorbent plate had a specific number of fins (16, 25, or 32) distributed at regular intervals and with equal dimensions in each case.

The performance of the PSS-IPRF was examined in relation to nanofluid mixed with a graphene and titanium dioxide nanoparticle composite (2.5 wt.% nanoparticles and 97.5% wt.% saline water). The nanoparticle properties are listed in Table 1.

**Table 1.** Real nanoparticle characteristics under investigation.

| Symbol | Density (g/cm$^3$) | Specific Heat (kJ/kg·K) | NPs Size (nm) | Conductivity (W/m K) |
|---|---|---|---|---|
| TiO$_2$ | ~4.05 | ~0.695 | 10–20 | ~11.8 |
| Graphene | ~2.267 | ~0.700 | 10–20 | ~4000 |

### 2.2. Experimental Procedure

The tests were conducted for the studied variables on both distillers simultaneously under the same weather conditions in Baghdad, Iraq (latitude 33° N, longitude 44° E). All experiments commenced after ensuring that the glass covers were clean at eight o'clock in the morning and continued until five o'clock in the evening between February and May 2022. The solar radiation, air velocity, saltwater temperature, glass cover temperature, and freshwater per hour were recorded with the water depth inside the solar stills fixed at 3 cm using a constant head tank. The results were recorded for CPSS and PSS-IPRF first and then for CPSS and PSS-HCPF the next day to ensure similar atmospheric conditions. The experiments were repeated in cases of large differences in the amount of water produced from CPSS for both days. In addition, experiments were conducted to study the effect of varied numbers of fins on the absorption plate (16, 25, and 36 fins), as well as the effect of using nanofluids for aquarium water mixed with titanium oxide nanoparticles ($TiO_2$) and graphene on the pyramid solar still performance.

### 2.3. Measuring Devices

Eight thermocouples with a measurement accuracy of ±0.2 were inserted at different locations in the solar stills to measure the temperature of the brine in the distillation basin, glass, and outside perimeter. The magnetic field strength was measured using a solar irradiation meter (accuracy ±5 W). A measuring jar was used to determine the hourly productivity, and the distillate was weighed with an electronic balance. Table 2 shows the characteristics of the measuring devices.

**Table 2.** Characteristics of the measurement tools.

| Device | Parameter | Unit | Resolution | Accuracy | Range | Error |
|---|---|---|---|---|---|---|
| Solar power meter | Solar radiance | $W/m^2$ | $0.1\ W/m^2$ | $\pm 1 W/m^2$ | $0\text{--}5000 W/m^2$ | 1.6% |
| K-type thermocouple | Temperature | °C | 0.1 °C | ±0.5 °C | 0–100 °C | 1.3% |
| Anemometer | Air speed | m/s | 0.01 m/s | ±0.1 m/s | 0.4–30 m/s | 1.1% |
| Balance | Yield | kg | 0.01 kg | ±0.2 kg | 0–25 kg | 1.3% |

Error analyses were performed according to the Holman technique [72] as follows:

$$W_R = \sqrt{\left(\frac{\partial R}{\partial X_1}W_1\right)^2 + \left(\frac{\partial R}{\partial X_2}W_2\right)^2 + \ldots + \left(\frac{\partial R}{\partial X_n}W_n\right)^2}$$

where $W_1, W_2, W_3, \ldots \ldots .., Wn$ a are the uncertainties of the independent parameters. Table 2 contains all computed tool errors.

$$\eta_{th} = f\left(\dot{m}, I_R, \Delta T_{w-g}\right)$$

The efficiency uncertainty was then calculated as:

$$W_{\eta_{th}} = \left[\left(\frac{\partial \eta_{th}}{\partial m}W_m\right)^2 + \left(\frac{\partial \eta_{th}}{\partial I_R}\right)^2 + \left(\frac{\partial \eta_{th}}{\partial \Delta T_{w-g}}\right)^2\right]^{\frac{1}{2}}$$

## 3. Results and Discussion

### 3.1. Variation in Solar Intensity and Ambient Air Temperature

Figures 4 and 5 show the change in ambient temperature and solar radiation during daylight hours for different months of the year. The thermal behaviour is similar to the hourly change in solar radiation and air temperature; the temperature gradually rises starting in the morning and increases with increased solar radiation, reaching a maximum value of between 12.00 and 14.00, depending on the change in the intensity of solar radiation during daylight hours. The maximum air temperature and solar intensity for February,

March, April, and May are 15.3 °C and 795 w/m$^2$; 23.6 °C and 922 w/m$^2$; 29.4 °C and 1088 w/m$^2$; and 36 °C and 1290 w/m$^2$, respectively. After the maximum value of solar radiation, the temperature begins to gradually decrease due to the reduced intensity of solar energy, reaching zero in the evening.

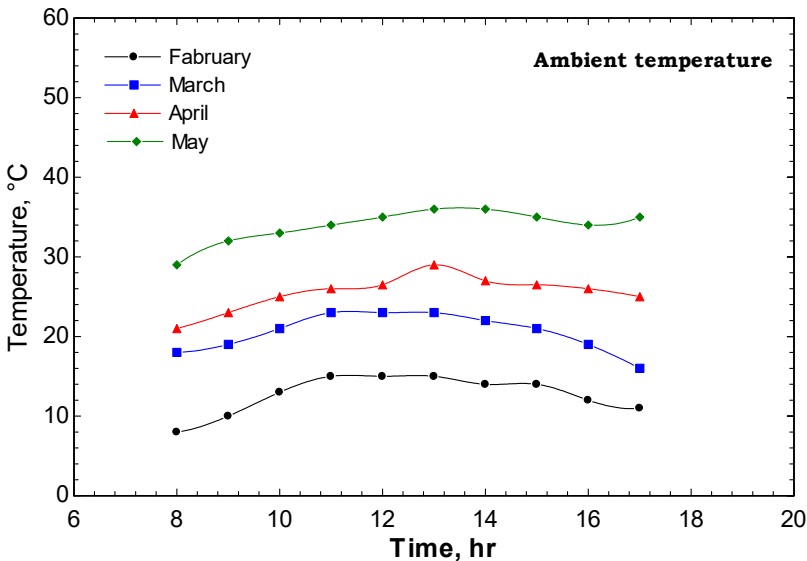

**Figure 4.** Hourly variations in air temperature for different months in 2022.

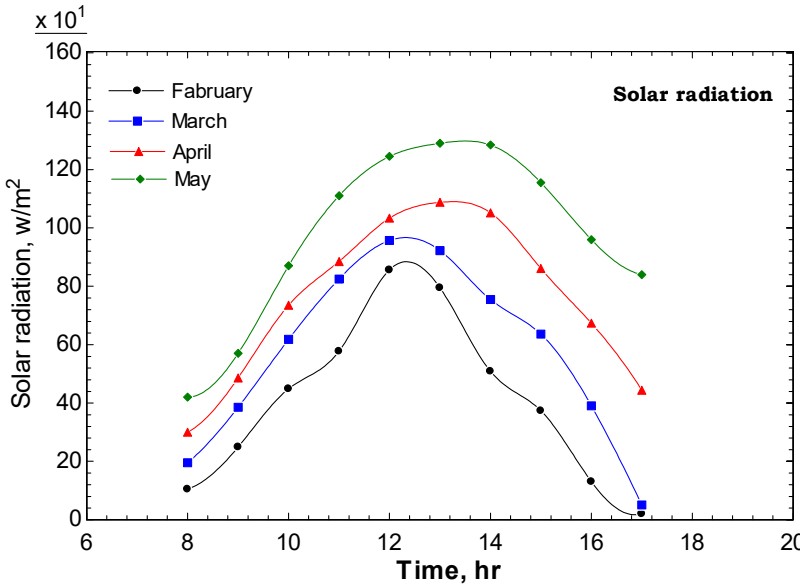

**Figure 5.** Hourly variations in solar irradiance for different months in 2022.

### 3.2. Performance of PSS-HCPF

Figure 6 shows the hourly changes in solar radiation and the temperature of the air, water, and glass of CPSS and PSS-HCPF, indicating that the water and glass temperatures of PSS-HCPF were higher than those of CPSS because the amount of vapour generated was greater in the PSS-HCPF than in CPSS. Moreover, the highest solar irradiance at 13.00 was 1100 W/m$^2$. The surface area of the water exposed to sunlight was 70 cm × 70 cm for CPSS and PSS-HCPF. PSS-HCPF had a higher water temperature compared to CPSS is because there was less aquarium water in PSS-HCPF than in the CPSS due to the mass of the cylindrical fins submerged in the basin still. Additionally, the bulk of the energy falling on the surfaces of the cylinders contributes to an increase in the temperature of the cylinders, accelerating evaporation because it improves the heat exchange between

the absorption plate and the water. In addition, the glass temperature was higher for PSS-HCPF due to the higher rate of evaporation and condensation, which resulted in a higher glass temperature.

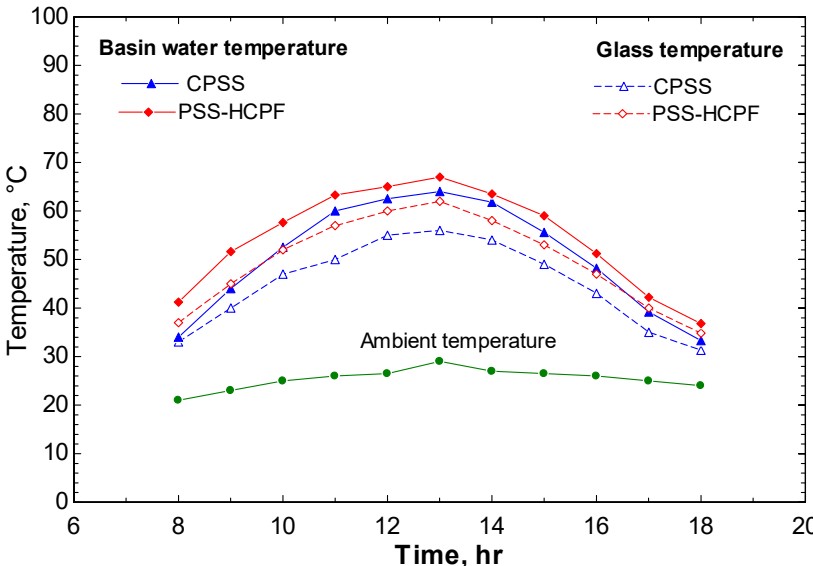

**Figure 6.** Hourly temperature variation for the CPSS and PSS-HCPF.

The hourly and total productivity of distilled water for both CPSS and PSS-HCPF are presented in Figure 7, showing that both stills have low productivity in the early morning, then gradually increase until midday, after which productivity begins to decrease with the decreased solar radiation. The maximum hourly productivity of CPSS was 490 mL/m$^2$·hr attained at 12.00, with a productivity of 610 mL/m$^2$/h per hour for PSS-HCPF and a maximum productivity of 13.00. The productivity of both stills decreased with decreasing solar radiation but decreased faster in conventional distillers. The daily yield of PSS-HCPF was 4840 mL/m$^2$ per day, whereas that of CPSS was 3718 mL/m$^2$ per day, indicating that the use of perforated cylindrical fins increased the yield by 31.3%.

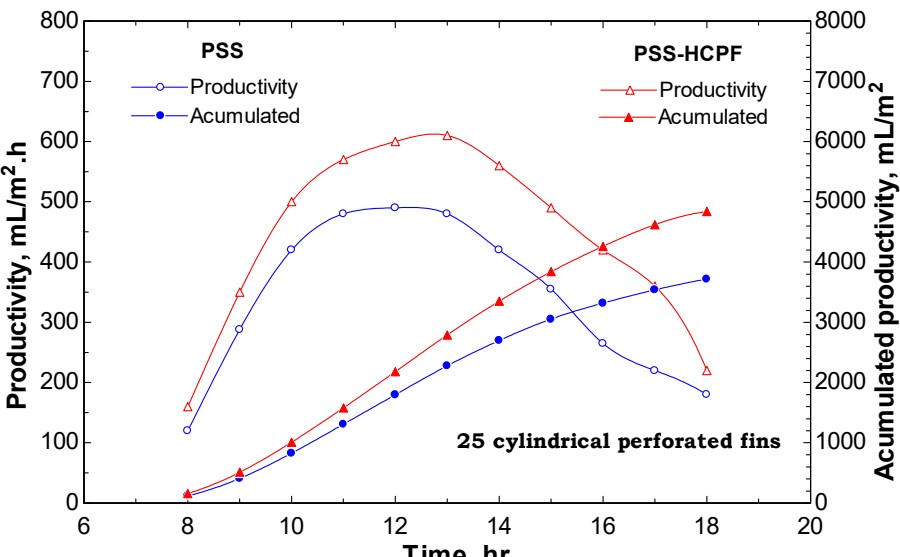

**Figure 7.** Hourly variation in productivity and overall accumulated productivity for the CPSS and PSS-HCPF.

### 3.3. Performance of PSS-IPRF

The hourly variation in the temperature of the basin water, cover glass, and the surrounding air for both solar stills, CPSS and PSS-IPRF, are illustrated in Figure 8. The PSS-IPRF had higher water and glass temperatures than the CPSS because the amount of vapour generated was greater in PSS-IPRF than in CPSS. The increase in water temperature is attributed to the improved thermal performance of the PSS-IPRF compared to CPSS, as the water temperature was considerably affected by the combination of the inclined perforated fins and the absorber plate. Moreover, the inclined perforated fins provided an increased absorption temperature of the absorption plate compared to the conventional still with a flat plate absorber.

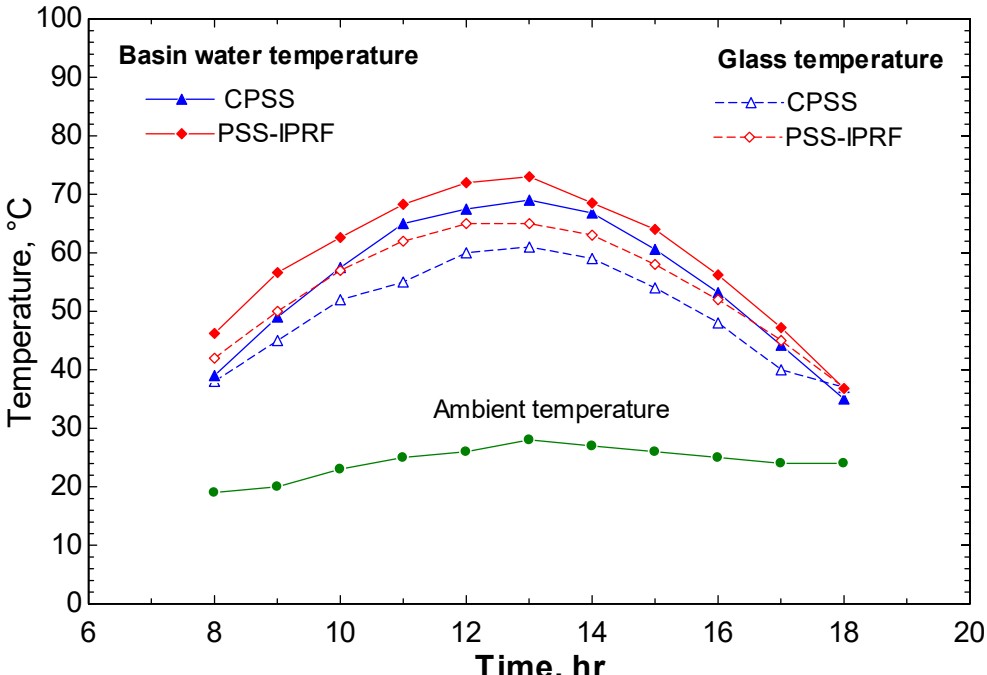

**Figure 8.** Hourly temperature variation for the CPSS and PSS-IRF.

Like the PSS-HCPF, the water depth was fixed at 3 cm, with 25 fins. The experiments were performed on successive days. The rate of solar radiation and wind speed was 1.6 m/s and 382 W/m$^2$, respectively. Figure 9 illustrates the hourly and total produced condensate water from PSS-IPRF and CPSS, showing that production of the PSS-IPRF begins before the CPSS because the flat inclined fins act as an additional endothermic surface, contributing to heating of the distillation tank more quickly compared to PSS and PSS-IPRF. For CPSS, the hourly productivity was 338 mL/m$^2$ h, and the maximum productivity of 490 mL/m$^2$ h was obtained at 12.00, whereas the hourly productivity of the PSS-IPRF was 451 mL/m$^2$ h, and the maximum productivity of 720 mL/m$^2$ h occurred at 13.00. After reaching maximum productivity, the productivity of both distillers decreased with decreasing solar radiation, with a faster decrease in productivity in the conventional distillers. The daily productivity of PSS-IPRF was 5750 mL/m$^2$—55.4% higher than the CPSS, indicating that the use of perforated planar fins with a perforated surface increases productivity.

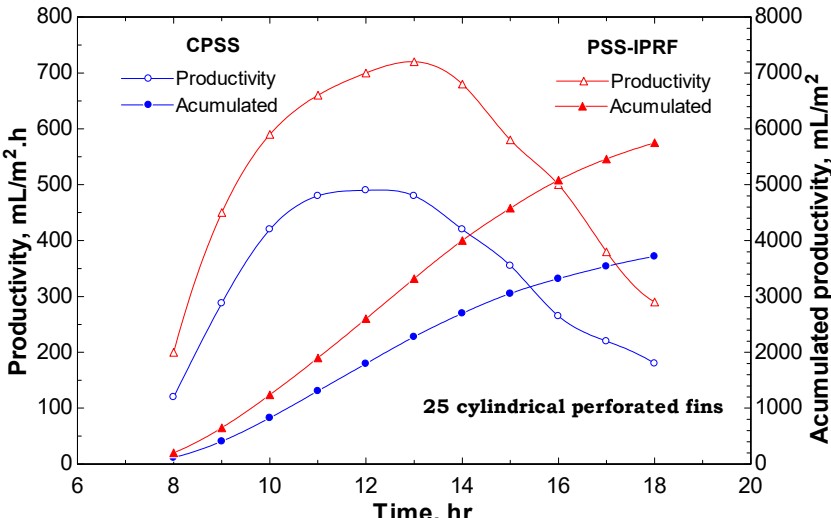

**Figure 9.** Hourly variation in productivity and overall accumulated productivity for the CPSS and PSS-IPRF.

*3.4. The Effect of the Number of Fins on the Daily Productivity of PSS-HCPF and PSS-IPRF*

Figure 10 presents the daily yield of the two solar stills using different numbers of fins, showing that increasing the number of fins from 16 to 36 fins led to reduced productivity; this can be explained as follows:

a. Initially, the PSS-HCPF with 16 fins achieved an increase in productivity of 21.6% compared to the CPSS, and increasing the number of fins to 25 further increased the daily productivity from 4480 mL/m$^2$ to 4843 mL/m$^2$. Increasing the number of fins to 36 further increased production by 41.1%.

b. Similar to the first case, the PSS-IPRF with 16 fins increased productivity by 38.8% compared to the CPSS, and increasing the number of fins to 25 increased daily productivity from 5120 mL/m$^2$ day to 5750 mL/m$^2$ day. Furthermore, increasing the number of fins from 25 to 36 increased the daily productivity from 5750 mL/m$^2$ day to 6070 mL/m$^2$ day, that is, the daily productivity increased by 64.5%. The flat fins inclined at an angle of 45 degrees served as an additional absorbing surface for solar radiation, in addition to improving the heat exchange between the absorbent plate and the water in the basin still.

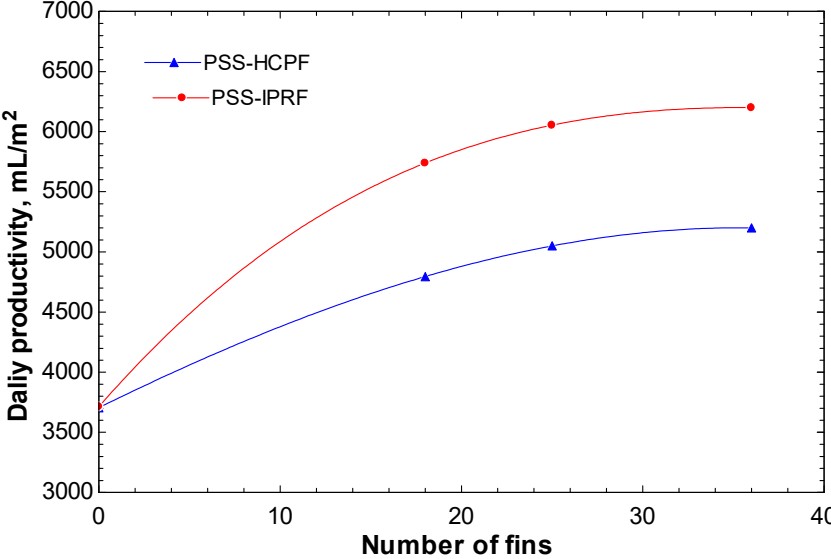

**Figure 10.** Effect of the number of fins on the daily productivity increase in PSS-HCPF and PSS-IPRF.

### 3.5. Comparisons between PSS-HCPF and PSS-IPRF Productivity

The use of a finned absorber plate, whether with cylindrical or inclined rectangular fins, improved the thermal performance of the modified solar stills because the fins increased the surface area for heat transfer between the absorption plate and the basin water, in addition to reducing the amount of water inside the distillation basin due to the space occupied by the fins, thereby increasing evaporation and productivity. Figure 11 provides a comparison of the cumulative yield versus time of the three solar stills—PSSC, PSS-HCPF, and PSS-IPRF—with 25 fins, showing that PSS-IPRF has the highest productivity because the rectangular inclined fins contribute significantly to the additional heating of the water due to the inclination angle of the fins. The inclination angle of the fins (45°) represents the optimum angle for absorption of solar radiation on a flat surface during the winter season in the Iraq region. In addition to increasing the surface area for heat transfer between the distilled basin and the water, the daily cumulative production of the three solar stills was 3720, 4850, and 5760 mL/m$^2$, which translates to an improvement in productivity over traditional solar stills of 31% for the cylindrical finned sun still and 56% for the flat inclined-fin solar still.

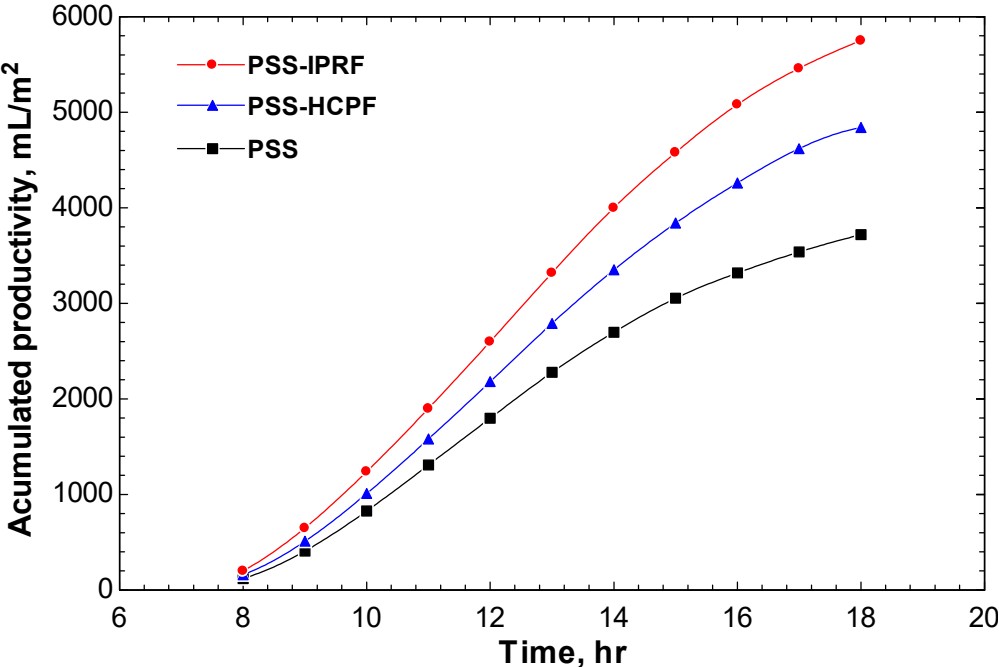

**Figure 11.** Productivity comparison between PSS-HCPF and PSS-IPRF.

### 3.6. Performance of PSS-IPRF with Nanocomposite

The performance of the PSS-IPRF was then investigated using TiO$_2$ and graphene nanocomposites, showing that the nanocomposites raised the water temperature of PSS-IPRF by 1–10 °C compared to CPSS. This indicates that the water temperature was elevated by the nanocomposites by around 1 °C. Additionally, the nanocomposites increased the glass temperature of the PSS-IPRF by 0 to 8 °C more than that of CPSS, indicating that the nanocomposites increased the glass temperature by around 1 °C. The daily productivity of the PSS-IPRF also increased by 82.1% compared to the CPSS, as shown in Table 3. Moreover, a comparison of the present results with those reported in previous studies (Table 4) confirms the viability of the proposed modified solar still.

**Table 3.** Water and glass temperature measurements, productivity, and productivity increase for PPS-IPRF with and without nanocomposites.

| Parameter | Without Nanocomposites | With Nanocomposites |
|---|---|---|
| Water temperature | PSS-IPRF above CPSS: 2 to 9 °C | From 1 to 10 °C for PSS-IPRF over CPSS |
| Glass temperature | PSS-IPRF above CPSS: 0 to 7 °C | From 0 to 8 °C for PSS-IPRF over CPSS |
| Daily productivity | PSS-IPRF: 6070 mL/m$^2$/day; CPSS: 3718 mL/m$^2$/day | PSS-IPRF: 6780 mL/m$^2$/day; CPSS: 3722 mL/m$^2$/day |
| Productivity Improvement | 63.20% | 82.10% |

**Table 4.** A comparison of the results obtained in the present study with those reported in previous studies with respect to rotational speed and productivity increase.

| No. | Authors and Reference | Solar Still Type | Additions | Productivity Rise, % |
|---|---|---|---|---|
| 1 | Omara et al. [32] | Pyramid solar still | Convex dish absorbers and wicks | 78% |
| 2 | Essa et. al. [73] | Tubular solar still | Wicks | 175 |
| 3 | Alawee et al. [61] | Pyramid distiller | Dangled cords with baffles within compartments | 176% |
| 4 | Farouk et al. [54] | Pyramid distiller | Titanium oxide (TiO$_2$), aluminum oxide (Al$_2$O$_3$) and copper oxide (Cu$_2$O) | 36% 46% 57% |
| 5 | Alawee et al. [74] | Pyramid solar still | Cords of jute Cords of cotton | 122% 118% |
| 6 | Alawee et al. [24] | Pyramid solar still | Reflectors, cooling, and wick cords | 195% |
| 7 | Asadabadi and Sheikholeslami [75] | Pyramid solar still | Copper fins and insulation | 62.5% |
| 8 | Ghandourah et al. [76] | Pyramid solar still | Corrugated absorber | 52.54% |
| 9 | Present work | Pyramid solar still | Hollow cylindrical perforated fins (PSS-HCPF). Inclined rectangular perforated fins (PSS-IPRF). PSS-IPRF with nanocomposites. | 31.3% 55.9% 82.1% |

Many researchers have studied the effect of different nanocomposites on the performance of pyramidal solar stills. Farouk et al. [54] concluded that the average daily productivity of PSS with Cu2O, Al$_2$O$_3$, and TiO$_2$ at a nanoconcentration of 0.3% increased by 57%, 46%, and 36%, respectively, relative to a conventional PSS. The performance of a convex PSS was tested by Omara et al. [32], who painted the absorber with black paint mixed with titanium oxide (TiO$_2$), copper oxide (CuO), and silver (Ag) nanocomposites. The silver paint achieved a 24% increase in productivity over conventional PSS, whereas the effect of CuO and TiO$_2$ paint only increased daily productivity by 19% and 16%, respectively.

### *3.7. Cost Analyses*

Table 5 provides information on fixed costs for both the CPSS and PSS-IPRF with nanocomposites. Table 6 lists the presumptions and estimates for a few factors used in the economic analysis, including the system lifetime, the number of working days in a year, and interest rates. The formulae in Table 7 were used to calculate costs of desalination. Based on this information, the costs of producing desalinated freshwater by the CPSS and PSS-IPRF with nanocomposites were USD 0.029 and 0.026/L, respectively.

**Table 5.** Fixed costs of CPSS and PSS-IPRF with nanocomposites.

| Unit | CPSS (USD) | PSS-IPRF-Nano (USD) |
|---|---|---|
| Iron sheet | 25 | 25 |
| Fins | - | 20 |
| Glass | 20 | 20 |
| Support legs and ducts | 25 | 25 |
| Production | 25 | 40 |
| Paint | 10 | 20 |
| Nanoparticles | – | 30 |
| Total fixed cost (F) | 105 | 180 |

**Table 6.** Assumptions and estimations used in the economic analysis.

| No. | Variable | Mean | Value | Unit |
|---|---|---|---|---|
| 1. | $n$ | System lifetime | 20 | Years |
| 2. | $i$ | Interest rate | 15 | % |
| 3. | $N$ | Working days of year | 340 | Day |
| 4. | $F$ | System fixed cost | 180 for PSS-IPRF-Nano 105 for CPSS | USD |
| 5. | $M$ | Average yearly productivity | 2000 for PSS-IPRF-Nano 1080 for CPSS | L/m$^2$.year |
| 6 | $CPL$ | Costs of the desalinated freshwater | 0.026 for PSS-IPRF-Nano 0.029 for CPSS | USD |

**Table 7.** Economic analysis [77].

| No. | Relation | Description |
|---|---|---|
| 1. | $CRF = \frac{i\,(1+i)^n}{(1+i)^n - 1}$ | Capital recovery factor |
| 2. | $FAC = F\,(CRF)$ | Fixed annual cost |
| 3. | $SFF = \frac{i}{(1+i)^n - 1}$ | Sinking fund factor |
| 4. | $S = 0.2\,F$ | Salvage value |
| 5. | $ASV = S\,(SFF)$ | Annual salvage value |
| 6. | $AMC = 0.15\,(FAC)$ | Annual maintenance costs |
| 7. | $TAC = FAC + AMC - ASV$ | Total annual cost |
| 8. | $CPL = TAC/M$ | Cost of distilled water |

## 4. Conclusions

The use of hollow cylindrical perforated fins and inclined perforated rectangular fins (IPRF) increased the productivity of a conventional pyramidal solar still (CPSS). The productivity of the PSS-IPRF was further enhanced by using nanocomposites; thus, the proposed modified solar still is a simple, cost-effective method for desalinating sea water.

**Author Contributions:** Conceptualization, S.A.M. and A.B.; methodology, Z.M.O.; software, W.H.A.; validation, H.A.D., F.A.E. and A.S.A.; formal analysis, H.S.M.; investigation, I.A.; resources, W.N.R.W.I.; data curation, S.A.M. and A.B.; writing—original draft preparation, A.A.A.-A.; writing—review and editing, A.A.A.-A.; visualization, Z.M.O.; supervision, W.H.A.; project administration, H.A.D., F.A.E. and A.S.A.; funding acquisition, W.N.R.W.I. All authors have read and agreed to the published version of the manuscript.

**Funding:** This work was partially funded by UKM under research code GUP-2020-012.

**Data Availability Statement:** Not applicable.

**Acknowledgments:** We acknowledge UKM and UOT for their support.

**Conflicts of Interest:** The authors declare no conflict of interest.

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
