# Peer review of "Pyramidal Solar Stills via Hollow Cylindrical Perforated Fins, Inclined Rectangular Perforated Fins, and Nanocomposites: An Experimental Investigation"

_sustainability, doi:10.3390/su142114116_

Round 1

Reviewer 1 Report (Previous Reviewer 1)

English needs revision.

The authors have well justified the comments of reviewer. This manuscript can be accepted now.

Author Response

Thank you

Reviewer 2 Report (Previous Reviewer 2)

The authors have tried to answer all the queries and incorporated appropriate corrections in the manuscript as per the suggestion. However, plagiarism has been ignored. Plagiarism of the manuscript has been checked via Turnitin and it shows high plagiarism. The authors are requested to look for the same.

Author Response

Dear revierwer,

Thank you for your comment

Processed, manuscript plagiarism verified by Turnitin and showing a low rate of plagiarism

Thank you

Best regards

Reviewer 3 Report (Previous Reviewer 3)

Accept

Author Response

Thank you

This manuscript is a resubmission of an earlier submission. The following is a list of the peer review reports and author responses from that submission.

Round 1

Reviewer 1 Report

1. Language should be revised

2. Introduction needs some more strong and in depth discussion on latest designs of solar stills as given in 

a. A thermodynamic review on solar stills, (2022) Solar Energy, etc.

3. As shown in experimental setup, a transparent long tube is used, authors must focused on this and explain the direct radiative heat transfer effect on this channel for loss analysis of water.

4. Heat transfer analysis is missing

5. Uncertainty analysis is missing

6. How are the results validated? what about thermal modeling? 

7. A complete cost analysis is required

Reviewer 2 Report

Authors have studied the pyramidal solar stills using different fins and nanocomposites. The title of the paper is not self-explaining the work also, the abstract does not clearly state the work done. Further comments can be found below:

The English grammar of the manuscript is inferior. It should be revised by an expert.

Many typo errors are observed throughout the paper such as the y-axis in figure 4, the use of section 3.4 in the start of section 3.2, etc. In-depth revision is required.

Many similar works are present on the web of science, the authors have ignored those works. More recent papers need to added in the references and also need to be discussed in the introduction section to show the novelty of the work.

The authors have not discussed how they have used composites and nanoparticles, what were their sizes, methods of coating, etc.

The authors have mentioned that they have studied the variation of number of fins but the results do not show it properly.

Most important, the abbreviation of hollow cylindrical perforated fins should be HCPF rather than HPCF.

Reviewer 3 Report

Detailed comments:

1.      The English of the text should be checked

2.      The novelty of manuscript is totally missing

3.      In the Introduction part, correct TiO2 with TiO2, and, also in the all manuscript. Unit of measure must be corrected. Information about TiO2 and graphene must be included (e.g. advantages and disadvantages, properties, applications).  The following references can be included in the Introduction part to improve the quality of manuscript, because they provide relevant information:

ü  Titanium dioxide nanomaterials for photovoltaic applications, Chem. Rev. 114 (2014) 10095–10130, http://doi:10.1021/cr400606n.

ü  Antireflective coating based on TiO2 nanoparticles modified with coupling agents via acid-catalyzed sol-gel method, Applied Surface Science, 487, 2019, p. 819-824

ü  Simulation and fabrication of SiO2/graded-index TiO2 antireflection coating for triple-junction GaAs solar cells by using the hybrid deposition process, Thin Solid Films 570 (2014) 585–590 http://doi:10.1016/j.tsf.2014.05.024.

Author write: “Several researchers have studied the technique of using fins in a distillation basin.” – the researchers and technique must be indicated

4.      At photo indicated in Figure 1 must be indicate what represents all component/tools

5.      The state-of-art is totally missing from the manuscript. The author should have added the previous research outputs (conditions/parameter/overall results) by comparing the present one in tabular form to show the viability of the present study (only indicate the no. of References it is enough).

6.      All equipment and tools used in this study should be described in detail or further information should be provided (manufacturer, type, operational conditions, etc).

7.      For all chemical, reagents, materials must be indicated manufacturer, amount, concentration

8.      Correct ml with mL

9.      Correct at figure 4, Ambint with Ambient; and, also must be indicated the value of temperature. Air temperature or ambient temperature?

10.  For Figure 5, at legend Solar radiation or solar irradiance?

11.  At Figures 6 and 8, correct temperature with temperature

12.  Comparison between the obtained results and measured in this study with other reported studies should be done and included for more clarity (indicate values not just number of reference).

13.  Kindly cite 3-4 relevant articles from “Sustainability”, MDPI in the Introduction part.

14.  At Conclusions, must be included applications of pyramidal solar with other materials/nanoparticles

15.  The References are very old. Some could be replaced with new, re-edited books. Multiple References must be included. The manuscript must contain the relevant information to be attractive for readers (researchers), because science has advanced, and the information indicated in the manuscript is no longer valid. This part should include observed information, noted in the last 10-12 years. 

16.  Use the Journal template, both for writing the manuscript and for References